# Polymers and Nanoparticles for Statin Delivery: Current Use and Future Perspectives in Cardiovascular Disease

**DOI:** 10.3390/polym13050711

**Published:** 2021-02-26

**Authors:** Antonio Nenna, Francesco Nappi, Domenico Larobina, Emanuele Verghi, Massimo Chello, Luigi Ambrosio

**Affiliations:** 1Cardiovascular Surgery, Università Campus Bio-Medico di Roma, 00128 Rome, Italy; a.nenna@unicampus.it (A.N.); e.verghi@unicampus.it (E.V.); m.chello@unicampus.it (M.C.); 2Cardiac Surgery, Centre Cardiologique du Nord de Saint-Denis, 93200 Paris, France; 3Institute for Polymers, Composites and Biomaterials, National Research Council of Italy, 00128 Rome, Italy; domenico.larobina@cnr.it (D.L.); luigi.ambrosio@cnr.it (L.A.)

**Keywords:** statin, polymer, nanoparticle, coronary artery disease, inflammation, stent, regeneration, endothelial dysfunction

## Abstract

Atherosclerosis-related coronary artery disease (CAD) is one of the leading sources of mortality and morbidity in the world. Primary and secondary prevention appear crucial to reduce CAD-related complications. In this scenario, statin treatment was shown to be clinically effective in the reduction of adverse events, but systemic administration provides suboptimal results. As an attempt to improve bioavailability and effectiveness, polymers and nanoparticles for statin delivery were recently investigated. Polymers and nanoparticles can help statin delivery and their effects by increasing oral bioavailability or enhancing target-specific interaction, leading to reduced vascular endothelial dysfunction, reduced intimal hyperplasia, reduced ischemia-reperfusion injury, increased cardiac regeneration, positive remodeling in the extracellular matrix, reduced neointimal growth and increased re-endothelization. Moreover, some innovative aspects described in other cardiovascular fields could be translated into the CAD scenario. Recent preclinical studies are underlining the effect of statins in the stimulation and differentiation of endogenous cardiac stem cells, as well as in targeting of local adverse conditions implicated in atherosclerosis, and statin delivery through poly-lactic-co-glycolic acid (PLGA) appears the most promising aspect of current research to enhance drug activity. The present review intends to summarize the current evidence about polymers and nanoparticles for statin delivery in the field of cardiovascular disease, trying to shed light on this topic and identify new avenues for future studies.

## 1. Introduction

Atherosclerosis is considered a systemic chronic inflammatory disease [1], in which oxidized low-density lipoproteins (oxLDL) impair endothelial function, facilitating the penetration of LDL below the endothelial layer and inducing the expression of surface chemotactic proteins (selectins and adhesion molecules); this leads to accumulation of lipids in the tunica media of the arteries [2,3]. In parallel, the recruitment, accumulation and activation of monocytes with their differentiation into macrophages and foam cells enhances the inflammatory process: reactive oxygen species (ROS) lead to the conversion of LDL into oxLDL [3]. After those early stages of plaque formation and atherosclerosis, LDL and immune cells induce a self-enhanced mechanism that results in plaque progression and stabilization, with a necrotic core stimulating neovascularization and tissue apoptosis [4,5]. Endothelial damage or intraplaque catabolism (from matrix metalloproteinases of inflammatory cells) lead to plaque enlargement, plaque rupture and thrombotic events [2,3], clinically resulting in coronary artery disease (CAD).

Mechanisms involved in the late phases of the atheroma are complex, but lipid metabolism is known to have a pivotal role [3,6]. Statin treatment, with its lipid-lowering effect and its pleiotropic activity, is generally used in clinical practice through systemic administration, but systemic side effects (mainly myopathy, liver damage, digestive system problems or low blood platelet count) and low concentration in the target atheroma represent major issues that should be addressed to improve clinical outcomes [7]. Strategies that locally and specifically target the lipid component and the inflammatory burden of atheroma could overcome the present limitations. Nanomedicine and polymer-based targeted administration can sustain drug release over time; therefore, drug accumulation and distribution can be guaranteed to target tissues, thus maximizing the therapeutic effect with minimal systemic adverse effects [2,8,9,10,11]. Targeted drug delivery through polymers and nanoparticles, initially investigated in cancer research, might be extremely useful even in cardiovascular disease. The present review intends to summarize the current evidence about polymers and nanoparticles for statin delivery in the field of cardiovascular disease, with a comprehensive update of recently published work, trying to shed light on this topic and identify new avenues for future studies.

## 2. Methods

For this review, we evaluated all clinical and preclinical studies investigating the effects of statin on coronary artery disease through polymer-based or nanoparticle-based administration. Literature search through PubMed, Embase, EBSCO, Cochrane database of systematic reviews, and Web of Science was updated to February 2020, using the following search keywords in various combinations: (“Hydroxymethylglutaryl-CoA Reductase Inhibitors”(Mesh) OR statin OR statins OR atorvastatin OR rosuvastatin OR lovastatin OR pravastatin OR simvastatin OR *statin) AND (“polymer” (Mesh) OR “nanoparticle” (Mesh) OR “particle” (Mesh)) AND (“atherosclerosis” (Mesh) OR “coronary artery disease” (Mesh)). References of relevant articles were also checked for significant contributions.

## 3. Results

Polymer-drug conjugates, with their amphiphilic properties, are made of an outer dense biocompatible shell with an inner drug-incorporated core, and can be administered orally [12]. The outer shell reduced absorption through the gastrointestinal tract, thus prolonging the concentration in the blood circulation with increased control of drug delivery. In case of nonoral administration, the outer shell prevents enzymatic and nonenzymatic biodegradation and metabolism, increasing the concentration in the effector site [13].

Polymers and nanoparticles can help statin delivery and their effects in multiple ways in patients with CAD (Figure 1):increasing oral bioavailabilityenhancing target-specific interaction, leading to improved statin effects in terms of:
oreduced vascular endothelial dysfunction;oreduced intimal hyperplasia;oreduced ischemia-reperfusion injury;oincreased cardiac regeneration;opositive remodeling of the extracellular matrix;oreduced neointimal growth and increased re-endothelization.

Moreover, some innovative aspects described in other cardiovascular fields could be translated into the CAD scenario.

### 3.1. Increase Oral Bioavailability

Statins are commercially available as oral tablets. However, oral administration is hampered by an extremely low aqueous solubility, rapid metabolism and low bioavailability that can impair most of their clinical effect [14]. In fact, bioavailability of most commonly prescribes statins ranges between 5% (simvastatin) and 20% (rosuvastatin), with a bioavailability of 12% for the most commonly prescribed compound (atorvastatin) [15,16,17], attributed to their poor aqueous solubility [15,16], (Table 1).

As an attempt to improve statin bioavailability, a wide variety of polymer-based approaches were described in the literature, summarized in Table 2 and Figure 2.

Strategies aimed at improving aqueous solubility [17,18,19,20,21,22,23], or prolonging the time on the gastrointestinal tract [24,25,26], were gradually replaced by approaches that use the lymphatic system for systemic absorption [16,27]. In fact, some nanoparticles were shown to be able to reach the lymphatic circulation of the intestine (M-cells and Peyer’s patches) [28,29,30], thus avoiding the enzymes of the enterocytes and the hepatic first-pass effect [16,31]. Cellulose-based polymers [16], were found to be superior [32,33], compared to cellulose-free polymers [27,28,29,30], because they guarantee a sustained release of the drug [32], due to better mucoadhesive [34], mechanical [35] and film-forming [35] properties and enhanced stability among physiologic pH levels [16,17,32]. Among cellulose-free polymers, compounds with poly-oxy-propylene (hydrophobic fraction) and poly-oxy-ethylene (hydrophilic fraction) induced a 4-fold increase in the oral bioavailability of atorvastatin [17], but the intrinsic variability of the compounds in terms of dimension and composition should be carefully investigated in tailored studies. Ethylcellulose combines the benefits of biodegradability and biocompatibility with a convenient cost-to-benefit ratio for large scale application [32,34], and is the most promising biocompatible polymer to act as a carrier for oral statin. In a recent study with a preclinical setting [16], atorvastatin-loaded ethylcellulose nanoparticles (with dimensions of 140–450 nm and entrapment efficiency of 65–85%) were found to be safe and effective in improving oral bioavailability with a 3.5-fold increase, suggesting that clinical application of ethylcellulose nanoparticles awaits in the near future.

### 3.2. Reduce Vascular Endothelial Dysfunction

Endothelial dysfunction is one of the key aspect of atherosclerosis [6,36,37], and its treatment remains debated and difficult due to intrinsic difficulties in targeting compounds into the endothelial cells. In this setting, nanoparticles able to target selectively the endothelium through specific endocytic pathways updated the therapeutic possibilities [38,39]. Vascular endothelial dysfunction is the final common pathway of many transduction systems (Table 3) [36,40,41,42], ultimately resulting in alteration of endothelial nitric oxide synthase, nitric oxide production and unbalanced oxidative stress [43].

Also, endothelial dysfunction is associated with aging and many risk factors for cardiovascular disease such as diabetes, smoking and obesity [6,43,44]. In a cellular study, pravastatin-loaded vesicles of poly-di-methyl-siloxane and poly-2-methyl-oxazoline were used to reduce the oxidative and inflammatory burden through reduced activation of macrophages [45], significantly implicated in endothelial dysfunction and rupture of high risk atherosclerotic plaques [1,2,3,45]. Nanoparticle-based delivery of simvastatin was also shown to reduce proliferation of macrophages in atherosclerotic plaques in case of advanced disease [46,47], with reduction of the inflammatory milieu and initiation of a positive remodeling [47], through the inhibition of the mevalonate pathway [46]. Similarly, pitavastatin-loaded poly-lactic-co-glycolic acid (PLGA) inhibited monocyte recruitment via underexpression of chemoattractant proteins and their stimulating factors, thus reducing plaque destabilization and rupture [48]. If sustained over time [46,47,48], the selective inhibition of macrophage activity might represent a potential target to treat inflammation in atherosclerosis. A recent study attempted to perform a superselective atorvastatin delivery to activated macrophages, by using macrophage membrane coated in reactive oxygen species (ROS)-responsive nanoparticles [49]. This system escapes clearance from the reticuloendothelial system and delivers nanoparticles to activated (“ROS-responsive”) inflammatory cells [49]. Targeted atorvastatin reduced plaque vulnerability due to increased collagen, decreased matrix metalloproteinases, decreased perivascular neovessel, decreased Ki67-related endothelial proliferation and optimized lipid profile; on the other hand, this innovative nanoparticle scavenged proinflammatory products leading to suppression of inflammation, increasing the therapeutic efficacy [49]. This nanoparticle-based biomimetic approach, based on ROS responsiveness, might be an interesting option for future studies as the combination of immune cell membrane coating and specific targets might actively modify the pathologic microenvironment and improve vascular endothelial function.

### 3.3. Reduce Intimal Hyperplasia

A subsequent aspect of vascular endothelial dysfunction is intimal hyperplasia, which represents a physiologic response to dangerous stimuli [50,51]. Common oral administration of statins is known to reduce intimal hyperplasia only by about 25% [52,53]. On the other hand, simvastatin-loaded nanoparticles (polysialic acid-polycaprolactone) were able to dramatically reduce vascular smooth muscle cells chemotaxis and intimal hyperplasia [52]. This antiproliferation signaling is related to nitric oxide blocking the Rho pathway, as statins act directly by increasing nitric oxide production and indirectly by lowering isoprenoids concentrations [52,54]. Targeted local delivery might enhance those factors, as well as the decrease in the inflammatory response that ultimately leads to intimal hyperplasia.

### 3.4. Reduce Ischemia-Reperfusion Injury

In the acute phase of CAD, ischemia-reperfusion injury plays a dramatic role in myocardial damage [55]. Early studies showed that statin treatment was associated with decrease in myocardial infarction size through the PI3K/Akt pathway and “reperfusion injury salvage kinase” (RISK) pathway [56], but this was not confirmed in preclinical and clinical studies because of insufficient compound targeted to the infarcted area in case of systemic (untargeted) administration [57,58]. Nanoparticle-based approaches were gradually developed to overcome this limitation, with promising results [59,60]. A bioabsorbable poly-lactic-co-glycolic acid (PLGA) nanoparticle is able to reach a tissue in which ischemia reperfusion occurs through increased vascular permeability [61], and in a preclinical model of acute myocardial infarction intravenous administration of these loaded nanoparticles allowed a targeted pitavastatin delivery in area of ischemia reperfusion injury [62]. This resulted in a reduction of the extension of myocardial infarction with improvement of left ventricular function, thus protecting the cardiac tissue from ischemia reperfusion damage [62]. The same group confirmed those results in larger animals, suggesting that nanoparticle delivery might overcome the previous limitations of statin treatment [63].

### 3.5. Increase Cardiac Regeneration

“Neo-cardiomyogenesis” after myocardial infarction allows a limited replacement of aged cells with new myocytes, thus mitigating the effect of ischemic conditions [6,64,65]. Statins are known to improve the regenerative potential of cardiac cells, preventing the age-dependent senescent phenotype [6,64,66], and this refueled scientific research. Recently, in a model of chronic myocardial infarction in rats, statins (pravastatin, simvastatin and rosuvastatin) enhanced resident cardiac stem/progenitor cell expansion with growth and differentiation towards the myogenic lineage [64].

Based on those suggestions, polymers and nanoparticles were used for those purposes and to confirm encouraging results of pioneering studies.

Yokoyama et al. [67], used a poly-lactic-co-glycolic acid (PLGA) nanoparticle to target simvastatin on the ischemic myocardium in mice. Simvastatin-loaded PLGA significantly enhanced cell migration and growth factor expression in vitro, and intravenous administration induced endogenous cardiac regeneration in their model of ischemic heart disease [67]. This regenerative ability was supposed to be related to adipose-derived stem cells, in which local simvastatin exerted a crucial effect. The granulation tissue of chronic infarction was gradually and largely replaced with regenerated myocardium after 1 month, and this was paralleled with a reduction of fibrosis and scar [67]. Adipose-derived stem cells are extremely appealing for future research because of their plasticity [68], their easy harvesting with minimally invasive procedures [67], and their ability to differentiate into cardiomyocytes and secrete angiogenic factors [69,70,71]. Polymer-based methods require fewer stem cells for transplantation, thus minimizing the risks of thromboembolism, without increasing statin adverse side effects related to systemic concentrations [67]. Additionally, this method is not extremely expensive because it employs somatic stem cells and commercially available compounds and is easily performed in preclinical settings [67].

Cardiac fibrosis, a common final pathway after chronic ischemic injury, is anticipated by a myofibroblast to fibroblast transdifferentiation due to impaired energetic supply [72,73]. Therefore, avoiding differentiation or promoting dedifferentiation are known targets to counteract cardiac fibrosis [73,74]. In this scenario, Emelyanova et al. [72], recently showed that the pleiotropic effect of statins encompasses this effect, and the underlying effect involves geranyl-geranyl pyrophosphate sensitive signaling. Further studies will show whether a polymer-based approach might be applicable to clinical studies.

Statins therefore might contribute to cardiomyocyte turnover after an ischemic injury, and statin-loaded polymers might represent a new therapeutic approach for CAD. If confirmed in human studies, similar effects might be responsible for the clinical effectiveness of statins in CAD.

### 3.6. Positive Remodeling in the Extracellular Matrix

The wound healing process after myocardial infarction and during coronary artery disease involves significant changes in the extracellular matrix, during the inflammatory, reparative and maturation phases [75]. Ultimately this leads to myofibroblast secretion of extracellular matrix to replace lost myocardial tissue with a definitive scar [76]. Mao et al. showed that, differently to routine pitavastatin administration, pitavastatin-loaded PLGA nanoparticles are able to protect the myocardial extracellular matrix from postischemic remodeling through the inhibition of monocyte mobilization from bone marrow [77], confirming previous suggestions and the hypothesis that targeted statins can improve the wound healing process after myocardial infarction or endothelial disruption [48,78,79,80]. Notably, PLGA scaffolds appear extremely interesting for statin delivery due to easy degradation, sustained drug release features and low probability of systemic reactions [79,81], and pitavastatin-PLGA was proposed as a new therapeutic strategy to cope with postischemic left ventricular remodeling in case of acute or chronic CAD79, and is being tested in some ongoing clinical trials [77]. Other studies reported atorvastatin-induced improvement of left ventricular remodeling (chamber volume and fibrotic content) after acute myocardial infarction through interaction with collagen metabolism (such as matrix metalloproteinases and their inhibitors) [82], and polymer-based approaches are awaited as a confirmation [83].

### 3.7. Reduce Neointimal Growth and Promote Re-Endothelialization

Percutaneous coronary interventions (PCI) with coronary stent implantation became a worldwide treatment for acute coronary syndrome and, in selected cases, for patients with chronic angina. Bioactive polymers in coronary stents became the mainstay of treatment to reduce complications, such as in-stent restenosis [84,85]. In-stent restenosis occurs in about 25% of patients undergoing PCI and is characterized by neointimal hyperplasia (vascular smooth muscle cells proliferation and migration, with the mTOR pathway playing a pivotal role [86]) with extracellular matrix adverse remodeling [85]. For this reason, bare-metal stents were gradually replaced by drug-eluting stents (such as everolimus and sirolimus), which showed a reduced risk of in-stent restenosis [84,85]. Statins were investigated to reduce those complications considering their beneficial effects on the endothelium, such as the inhibition of intimal hyperplasia/cell proliferation, the inhibition of the mTOR pathway via AMPK activation and the promotion of autophagy among macrophages reducing inflammation [86,87]. As mTOR mediates many of the pathological features of in-stent restenosis [85], statin-eluting stents might have profound clinical implications in PCI [86]. However, this topic was not adequately investigated in the recent literature, but pioneering studies produced significant and interesting results.

Miyauchi et al. [88], compared cerivastatin-eluting stents to bare-metal stents in an animal model, showing a decrease in inflammatory response and improved neointimal function. Similar results were confirmed in later studies [89,90], and direct comparison with sirolimus-eluting stents showed similar in-stent restenosis rates but with improved healing effects on the endothelium, thus theoretically reducing the risk of late thrombosis [91]. Although statin-eluting stents might prevent the endothelial proliferation of in-stent restenosis and promote re-endothelialization avoiding late thrombosis, their use in clinical studies remain limited probably due to economic constraints.

### 3.8. Translational Outlooks

Peripheral arterial disease (PAD) represents a noncardiac localization of the atherosclerotic disease. Considering that the atherosclerotic and inflammatory burden is similar between PAD and CAD [92], results from PAD studies could be theoretically translated into the CAD scenario. Although scientific validation is required before drawing definitive conclusions, it appears reasonable to use the suggestions from the “PAD world” to design similar studies in the “CAD world”.

Therapeutic angiogenesis is a milestone in PAD treatment, and reparative collateral growth was attempted over the years with early pioneering reports refuted by later studies [93]. Bioabsorbable poly-lactic-co-glycolic acid (PLGA) nanoparticles loaded with pitavastatin produced encouraging results in preclinical settings. In this scenario, intramuscular injection of pitavastatin-loaded PLGA improved blood perfusion, microvascular angiogenesis and macroscopic arteriogenesis after limb ischemia through pleiotropic mechanisms [7,59,60]. Also, the use of nanoparticle-based delivery increases the collateralization with a less than 10% cumulative dose compared to systemic administration [59,60]. Pitavastatin-loaded PLGA was used in a phase 1 clinical trial [93] in patients with PAD, but no specific results were reported.

Aortic aneurysm shares a proinflammatory condition with atherosclerosis [6,92]. A recent article evaluated an animal model of abdominal aortic aneurysm in which pitavastatin-loaded polymer (poly-ethylene-glycol poly-lysin-phenylboronic acid, PEG-PLysFPBA) dose-dependently reduced aneurysm expansion [12]. Also, the statin group showed a reduced macrophage infiltration and decrease activity of matrix metalloproteinases [12].

In the preclinical field of vascular surgery, the group of Liu et al. [94,95] performed interesting studies on models of carotid artery injury. Pitavastatin-loaded nanoparticles with PLGA improved endothelial progenitor cells proliferation and were more effective in repairing injured vasculature via the PI3K signaling pathway, by promoting re-endothelialization and reducing intimal hyperplasia [95]. Similarly, rosuvastatin-loaded nanoparticles with poly(L-lactide-co-caprolactone) (PLCL) favored endothelialization and reduced thrombotic potential through increased vascular endothelial growth factor (VEGF) signaling [94]. Those models can be extremely useful in the CAD setting, to counteract in-stent restenosis and improve long-term durability of coronary stents.

Other interesting reports were published about pulmonary artery hypertension [96]. In two studies, poliylactide-glycolide mediated pitavastatin delivery was able to reduce the progression of pulmonary artery hypertension in animal models through anti-inflammatory and antiproliferative effects [97,98].

Tuerdi et al. [99] showed that simvastatin-loaded nanoliposomes were able to counteract the negative cardiac remodeling in isoproterenol-induced cardiomyopathy, reducing hypertrophy, fibrosis and inflammation [99].

Those strategies appear reproducible and might be translated into the CAD scenario, and similar/clinical studies are awaited. The recent interest in the interaction between statins and platelets might be another powerful and interesting tool against CAD [100]. Targeted statin delivery, besides enhancing endothelial function, might reduce the progression of coronary plaques and their rupture with thrombosis through an additive effect with antiplatelet agents [100].

## 4. Conclusions

Advanced atherosclerotic plaques have specific features such as neovascularization, microcalcification, cholesterol accumulation and cellular necrosis that can lead to destabilization of the plaque and clinically significant cardiovascular events [1,2,101]. Those mechanisms are highly dependent on lipid content and local inflammatory events. Statin treatment represents a milestone, but the effects of systemic administration might be improved by targeted statin delivery through polymers and nanoparticles [9,10] (Table 4). Recent preclinical studies are underlining the effect of statins in the stimulation and differentiation of endogenous cardiac stem cells, as well as in targeting local adverse conditions implicated in atherosclerosis, and statin delivery through PLGA appears the most promising aspect of current research due to its benefit in different pathways (Figure 1, Figure 3). Future studies will strengthen this hypothesis and will elucidate the clinical role of polymer-based statin delivery in the reduction of cardiovascular events in patients with coronary artery disease.

## Figures and Tables

**Figure 1 polymers-13-00711-f001:**
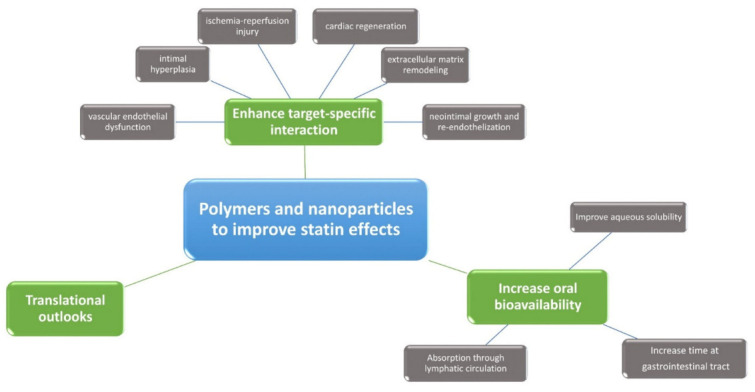
Polymers and nanoparticles for statin delivery: main effects.

**Figure 2 polymers-13-00711-f002:**
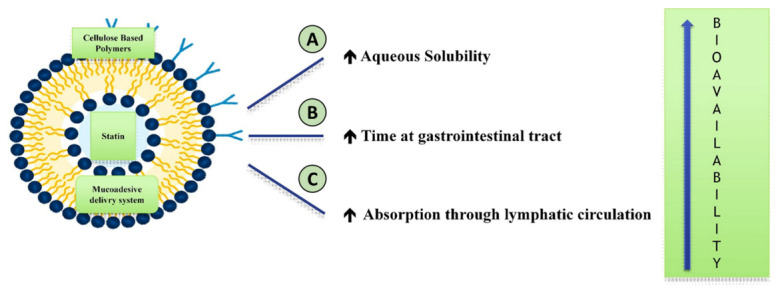
Structure of nanoparticles increasing the bioavailability of statins. Self-emulsifying vehicles and biodegradable inert polymers (A) (chitosan, hydroxy-propyl-merhilcellulose, polyvinylpyrrolidone, cyclodextrin) increases their aqueous solubility; mucoadhesive delivery system (B) increases the gastrointestinal transit time; cellulose-based polymers (ethylcellulose) or cellulose-free polymers (C) (polycaprolactone, polylactide-co-glycolide, polyoxypropylene and polyoxyethylene) manage to increase the affinity and absorption through the lymphatic circulation, so as to minimize the metabolism through the liver.

**Figure 3 polymers-13-00711-f003:**
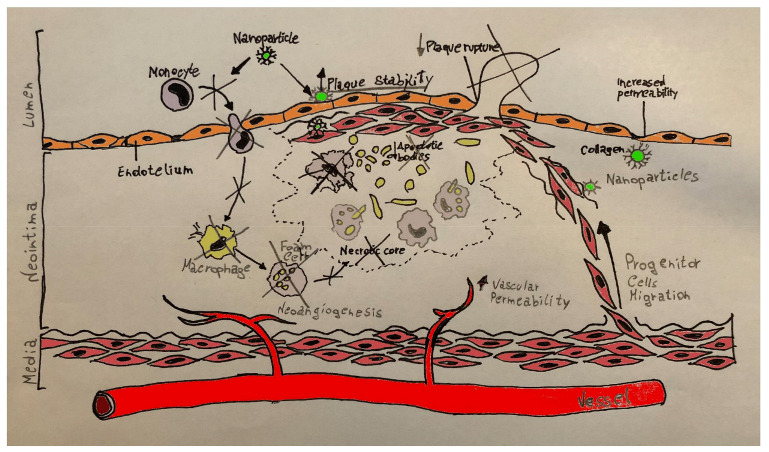
Poly-lactic-co-glycolic acid (PLGA) effects on atherosclerotic plaque and ischemic myocardium. PLGA promotes myocardial regeneration, plaque stability and reduces post-ischemia-reperfusion injury through different pathways. It inhibits the expression of chemoreceptor proteins and inhibits monocytes recruitment to prevent destabilization and rupture of the atherosclerotic plaque. By stimulating neoangiogenesis and increasing vascular permeability in tissues injured by ischemia it protects the tissue from damage from revascularization and myocardial fibrosis while preserving ejection fraction. By stimulating the production of tissue growth factors, it promotes cell migration of progenitor cells by promoting myocardial regeneration. By inhibiting the recruitment of monocytes from the bone marrow, it protects the myocardial extracellular matrix from postischemic remodeling.

**Table 1 polymers-13-00711-t001:** Pharmacokinetics features of commonly used statins (adapted from Askarizadeh et al. [14]).

Parameter	Atorvastatin	Simvastatin	Rosuvastatin	Pitavastatin	Cerivastatin	Fluvastatin	Lovastatin	Pravastatin
Prodrug	No	Yes	No	No	No	No	Yes	No
Hydrophilic	No	No	Yes	No	No	No	No	Yes
Fraction Absorbed (%)	30	70	Unknown	80	98	98	30	34
Bioavailability (%)	12	5	20	80	60	25	5	18
Active Metabolites	Yes	Yes	Yes	No	Yes	Yes	Yes	Yes
Half Time (hours)	15–30	2–3	20	11	2.5	0.5–2.5	3	1.5–2.5
Hepatic Metabolism (%)	70	78–87	63	Unknown	Unknown	70	70	45–65
Renal Metabolism (%)	2	13	10	2	30	6	30	60

**Table 2 polymers-13-00711-t002:** Potential methods to increase oral bioavailability of statins.

Rationale	Compound/Method	Ref.
Improve Aqueous Solubility	Modification of Particle Size	[18]
	Modification of Crystallinity	[19]
	Incorporation With a Self-emulsifying Vehicle	[17]
	Incorporation With Biodegradable Inert Polymer: Chitosan	[20]
	Incorporation With Biodegradable Inert Polymer: Hydroxypropyl Methylcellulose	[21]
	Incorporation With Biodegradable Inert Polymer: Polyvinyl Pyrrolidone	[22]
	Incorporation With Biodegradable Inert Polymer: Cyclodextrin	[23]
Increase Time at Gastrointestinal Tract (Retentive/Sustained Delivery Mechanisms)	Mucoadhesive Delivery System, Oral Mucosa	[24]
	Mucoadhesive Delivery System, Gastric Fluid	[25,26]
Absorption Through Lymphatic Circulation	Cellulose Free: Polycaprolactone	[27,28]
	Cellulose Free: Polylactide-co-glycolide	[29,30]
	Cellulose Free: Polyoxypropylene (Hydrophobic Fraction) and Polyoxyethylene (Hydrophilic Fraction)	[17]
	Cellulose-based: Ethylcellulose	[16]

**Table 3 polymers-13-00711-t003:** Signaling systems involved in endothelial dysfunction, potential targets for polymer-based approaches [36,40,41,42].

Factors *Increased* in Endothelial Dysfunction	Factors *Decreased* in Endothelial Dysfunction
Advanced Glycation End Products Caveolin Cholesteryl Ester Transfer Protein Endothelin Epoxide Hydrolase Geranyl-geranyl-transferase Janus Kinase Lipoprotein Lipase Poly-ADP Ribose Polymerase Protein Tyrosine Phosphatase Rho-kinase Transketolase	Angiotensin Converting Enzyme 2 Peroxisome Proliferator-activated Receptor Protein Kinase A Protein Kinase B Sphingosine Phosphate

**Table 4 polymers-13-00711-t004:** Summary of the effect of currently available statin-loaded nanoparticles.

PLGA	Poly-di-mehyl-siloxane Poly-2-2methyl-oxazoline	Nanoliposomes	Polysialic acid-polycaprolactone	PEG-PLysFPBA	PLCL
↓ Chemotactic proteins	↓ Macrophage activation	↓ Isoproterenol	↓ Nitric oxide	↓ Inflammation	↑ VEGF
↑ Post ischemic permeability	↓ Oxidative burden	↓ Fibrosis	↓ Rho pathway	↓ Cellular proliferation	
↑ Growth factors	↓ Inflammatory burden	↓ Inflammation			
↑ Micro/macro vascular angiogenesis					
↑ Endothelial progenitor cells proliferation					
↓ Monocyte mobilization					

## Data Availability

The data presented in this study are available on request from the corresponding author.

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
