# Peer review of "Polymers and Nanoparticles for Statin Delivery: Current Use and Future Perspectives in Cardiovascular Disease"

_polymers, 2021, doi:10.3390/polym13050711_

Round 1

Reviewer 1 Report

Polymers

polymers-1112790

Title: Polymers and nanoparticles for statin delivery: current use and future perspectives in cardiovascular disease

Article Type: Original Article

We are pleased to send you major comments. The topic of review was quite interesting. Overall, the presentation style of article is not acceptable for journal. I noticed that authors did not follow the journal instruction such as references setting, font, even the style. However, the authors should consider following comments for its acceptance for publication.

  1. In abstract, please highlight the novelty of the work.
  2. Line 65-68, the sentence should be rephrased, I did not get the meaning.
  3. The method section is very confusing, I did not get the meaning, what the authors want to describe?
  4. I am surprised, in review article only one figure?
  5. Overall, manuscript was written very weird and it not easy to understand at present form. Please revised throughout all, used proper data from literature to write review on this topic. There are many works reported already on this topic. Please mentioned clearly your novelty. The review is about to deliver an idea for others its not just rough collection of data.

Overall, I am suggesting the major revision. The present research cannot attract the readers.

Author Response

Reviewer 1

We are pleased to send you major comments. The topic of review was quite interesting. Overall, the presentation style of article is not acceptable for journal. I noticed that authors did not follow the journal instruction such as references setting, font, even the style.

Thank you for your comments. Manuscript presentation has been revised thoroughly according to journal’s template.

However, the authors should consider following comments for its acceptance for publication.

  1. In abstract, please highlight the novelty of the work.

Thank you for your comment. The importance of this work has been included at the end of the abstract. The present review intends to summarize the current evidences about polymers and nanoparticles for statin delivery in the field of cardiovascular disease.

  1. Line 65-68, the sentence should be rephrased, I did not get the meaning.

Thank you for your comment. The sentence has been changed: “Endothelial damage or intra-plaque catabolism (from matrix metalloproteinases of inflammatory cells) lead to plaque enlargement, plaque rupture and thrombotic events, clinically resulting in coronary artery disease (CAD).”

  1. The method section is very confusing, I did not get the meaning, what the authors want to describe?

Thank you for your comment. This section describes the methodology of the literature search for this manuscript (in terms of key words and updated information), like most review articles published in Polymers by our group. This allows reproducibility and improve clarity of presentation. The last sentence of the section has been removed to improve fluency.

  1. I am surprised, in review article only one figure?

Thank you for your comment. Three additional figures have been included in the manuscript, as suggested. The new Figure 2 summarized the concept of increased bioavailability. The new Figure 3 summarizes the different biological effects of different statin-loaded nanoparticles. The new Figure 4 summarized the effect of statin-loaded PLGA compounds.

  1. Overall, manuscript was written very weird and it not easy to understand at present form. Please revised throughout all, used proper data from literature to write review on this topic. There are many works reported already on this topic. Please mentioned clearly your novelty. The review is about to deliver an idea for others its not just rough collection of data.

Thank you for your comment. Manuscript has been revised in most sections as required.

Reviewer 2 Report

This reviewer recommends the manuscript to publish in Polymers as a present form.

Author Response

Reviewer 2

This reviewer recommends the manuscript to publish in Polymers as a present form.

Thank you for your comment and your appreciation.

Reviewer 3 Report

Polymers and nanoparticles for statin delivery: current use and future perspectives in cardiovascular disease is an interensting comprehensive review on statin delivery. The review is well written although some typos and grammar/spell check-up is necessary.

I have only minor comments that should help to improve the visibility of the manuscript

1) Authors have made only one scheme (Figure) and summarized all materials on tables. I do recommend to work on representation of the materials. You should better explain the approaches using Figures. You can draw your own Figs or contact the for copyrights

2) Please check for typos, e.g.

Line 57 E-selecttin instead of E-selection

Author Response

Reviewer 3

Polymers and nanoparticles for statin delivery: current use and future perspectives in cardiovascular disease is an interesting comprehensive review on statin delivery. The review is well written although some typos and grammar/spell check-up is necessary. I have only minor comments that should help to improve the visibility of the manuscript.

Thank you for your comments and your appreciation. English language has been revised throughout the manuscript and your comments have been revised accordingly.

1) Authors have made only one scheme (Figure) and summarized all materials on tables. I do recommend to work on representation of the materials. You should better explain the approaches using Figures. You can draw your own Figs or contact the for copyrights.

Thank you for your comment. Three additional figures have been included in the manuscript, as suggested. The new Figure 2 summarized the concept of increased bioavailability. The new Figure 3 summarizes the different biological effects of different statin-loaded nanoparticles. The new Figure 4 summarized the effect of statin-loaded PLGA compounds.

2) Please check for typos, e.g. Line 57 E-selecttin instead of E-selection

Thank you for your comment. Changes have been made accordingly.

Round 2

Reviewer 1 Report

I have reviewed again the manuscript and I found that the present form of the manuscript can be accepted for publication. The mostly concerns are well answered by authors in the revised version.